# Therapeutic Microbiology: The Role of *Bifidobacterium breve* as Food Supplement for the Prevention/Treatment of Paediatric Diseases

**DOI:** 10.3390/nu10111723

**Published:** 2018-11-10

**Authors:** Nicole Bozzi Cionci, Loredana Baffoni, Francesca Gaggìa, Diana Di Gioia

**Affiliations:** Department of Agricultural and Food Sciences (DISTAL), *Alma Mater Studiorum*—Università di Bologna, Viale Fanin 42, 40127 Bologna, Italy; nicole.bozzicionci@unibo.it (N.B.C.); loredana.baffoni@unibo.it (L.B.); francesca.gaggia@unibo.it (F.G.)

**Keywords:** *Bifidobacterium breve*, probiotics, paediatrics, therapeutic microbiology

## Abstract

The human intestinal microbiota, establishing a symbiotic relationship with the host, plays a significant role for human health. It is also well known that a disease status is frequently characterized by a dysbiotic condition of the gut microbiota. A probiotic treatment can represent an alternative therapy for enteric disorders and human pathologies not apparently linked to the gastrointestinal tract. Among bifidobacteria, strains of the species *Bifidobacterium breve* are widely used in paediatrics. *B. breve* is the dominant species in the gut of breast-fed infants and it has also been isolated from human milk. It has antimicrobial activity against human pathogens, it does not possess transmissible antibiotic resistance traits, it is not cytotoxic and it has immuno-stimulating abilities. This review describes the applications of *B. breve* strains mainly for the prevention/treatment of paediatric pathologies. The target pathologies range from widespread gut diseases, including diarrhoea and infant colics, to celiac disease, obesity, allergic and neurological disorders. Moreover, *B. breve* strains are used for the prevention of side infections in preterm newborns and during antibiotic treatments or chemotherapy. With this documentation, we hope to increase knowledge on this species to boost the interest in the emerging discipline known as “therapeutic microbiology”.

## 1. Introduction

The use of microorganisms to treat or prevent targeted diseases was conceived at the end of the last millennium. This concept has rapidly evolved giving rise to a new branch of applied microbiology known as “therapeutic microbiology” [1]. Since human organisms and gut microbiota establish an intimate symbiotic relationship that is fundamental for the maintenance of the host’s health, the administration of beneficial microorganisms may represent a key determinant of the general health status and diseases susceptibility. The choice for the most suitable species for a certain pathology requires extensive studies, both *in vitro* and *in vivo*. Moreover, it is known that strains belonging to the same species may express different functions *in vivo* [2]. It has also been demonstrated that blending different microbial strains, species or even genera, may lead to a final effect that is not predicted by results from using each single microorganism. Several *Bifidobacterium* species are largely used as probiotics for their capability of reaching and colonizing the gastrointestinal tract and their documented history of safety. Among them, *Bifidobacterium breve*, originally isolated from infant faeces, represents one of the most used probiotics in infants. The multiple studies in which *B. breve* strains have been successfully used in diseased humans, especially children and newborns, witness the potentiality of strains belonging to this species for the prevention or treatment of human diseases. The aim of this review is to show the various applications of *B. breve* for preventing and treating paediatric diseases starting from *in vitro* and mice model assessment of efficacy to the clinical use. To the best of our knowledge, this work represents the first collection of works focused on the application in paediatrics of strains belonging to the *B. breve* species and is aimed to shed light on the role of this *Bifidobacterium* species in the scenario of “therapeutic microbiology.” Moreover, this paper explores the effectiveness of *B. breve* used both as a single strain and combined with other microorganisms with a final short outcome of its application in adulthood.

## 2. The Human Intestinal Microbiota

The human intestinal microbiota is a complex ecosystem that includes not only bacteria but also fungi, Archaea, viruses and protozoans; bacteria concentration increases from the stomach and duodenum throughout the intestinal tract and in the large intestine it rises to 10^11^–10^12^ CFU/g of lumen content [3]. It has been estimated that at least 1800 genera and a range of 15,000–36,000 bacterial species, depending on whether species are conservatively (97% OTUs) or liberally (99% OTUs) classified, can be found in the large intestine [4].

The symbiotic mutualistic relationship that the gut microbiota establishes with the host exerts several beneficial roles, the main of which are the maintenance of the gut epithelial barrier, the inhibition of pathogen adhesion to intestinal surfaces, the modulation and proper maturation of the immune system, the degradation of otherwise non-digestible carbon sources such as plant polysaccharides and the production of different metabolites including vitamins and short chain fatty acids (SCFAs) [5]. Furthermore, intestinal microorganisms seem to be responsible for a bidirectional interaction between the gut and the Central Nervous System (CNS) via the gut-brain axis [6]. Dysfunction in this interaction may be implicated in the development and prognosis of some neurological diseases, including autism [7], multiple sclerosis [8] or Parkinson disease [9]. Because of this symbiotic relationship, the human organism can be seen as a “superorganism,” which consist of not only the microbial cells but also their genomes, that is, the microbiome and the related microproteome and micrometabolome [10]. The microbiome represents more than 100 times the human genome (1,000,000 genes vs. 23,000 genes) [10]. Indeed, the gut microbiome is influenced by external factors, such as diet, health status and xeno-metabolome. These factors shape the individual intestinal microbiota that can be considered as a “fingerprint” of the hosting organism.

Recently, the realization of global-collaborative projects has enriched the knowledge about the gut microbiota, such as the MetaHit project [11], the Human Microbiome project [12] and the MyNewGut project [13]. Moreover, the large amount of data from high throughput gene sequencing technology has allowed us to gain deeper insights in the composition of the “typical” human gut microbiota. The two principal bacterial phyla are *Firmicutes* and *Bacteroidetes*, followed by *Actinobacteria*, *Proteobacteria* and *Verrucomicrobia*. Fungi and Archaea constitute approximately 1% of the species of the intestinal microbiota [14,15]. The predominant fungal phyla are *Ascomycota* and *Basydiomicota*; some of the most abundant genera, that is, *Saccharomyces*, *Debaryomyces* and *Kluyveromyces*, are found in food, confirming the influence of diet habits also on the fungal intestinal population [16]. From the 80s some archaeal species belonging to *Methanobrevibacter* genus have been identified. *Methanobrevibacter* is the only genus detected in the gut probably due to the use of 16S primers not having sufficient resolution for Archaea. Within this genus, the species’ composition depends on diet and host’s health status, as for the entire microbiota [17,18].

Microbiologists’ attention has been also focused on bacteriophages, which, living at bacteria expense and being vehicles of genetic transfer, could have an important role in shaping the biodiversity of the gut ecosystem. The first metagenomic analysis of an uncultured viral community from human faeces using partial shotgun sequencing suggested a large diversity of phages in gut microbiota [19]. The same authors investigated the viral community in the infant intestine using metagenomic sequencing: 72% of the detected viral community resulted to be siphoviruses and prophages and over 25% resulted to be phages that infect lactic acid bacteria; faecal viral sequences were not identified in breast milk, suggesting a non-dietary initial source of viruses [20]. The entire viral community composition changed dramatically between the first and the second week of age [20], remaining then stable during host’s life [21].

Gut colonization begins at birth, although recent evidences suggest the existence of an intrauterine transmission of maternal bacteria to the foetus [22]. The first colonizer are facultative anaerobes (*Staphylococcus* spp., *Enterobacteriaceae* and *Streptococcus* spp.), followed by strict anaerobes, such as members of *Bifidobacterium*, *Bacteroides* and *Clostridium* genera [23,24]. The mode of delivery exerts a strong influence on the first microbial colonization of newborns’ gut. Children born by natural delivery have an intestinal microbiota profile similar to their mother’s vaginal one, characterized by *Lactobacillus* and *Prevotella* spp., while children born by caesarean section develop a microbiota similar to that of mother’s skin (*Streptococcus*, *Corynebacterium* and *Propionibacterium* spp.) [25]. In addition, the type of feeding has a crucial role on the colonization of microbial groups in the gut. Indeed, the gut microbiota of formula-fed infants contains a higher amount of *Escherichia*, *Veillonella*, *Enterococcus* and *Enterobacter* members and the concentration of *Lactobacillus* and *Bifidobacterium* is lower with respect to in breast-fed infants [26]. The abundance of these genera can be due to a more acidic pH in the colon of breast-fed infants [27]. The prevalence of bifidobacteria in breast-fed infants is also due to their capability of fermenting oligosaccharides (referred to as human milk oligosaccharides, HMO) [28]. Diet continues to exert a crucial influence in the gut microbiota composition also in adulthood: De Filippis et al. [29] showed an association between plant-based diet and a prevalence of *Lachnospira* and *Prevotella* and a positive correlation between *Ruminococcus* and omnivore diet. Animal-based diets increase the abundance of bile-tolerant microorganism (*Alistipes*, *Bilophila* and *Bacteroides*) and decreases the levels of *Firmicutes* [30].

The use of antibiotics influences the gut microbiota composition, determining a significant decrease of the microbial diversity in the digestive tract [31,32]. However, the microbiota is a resilient system and tends to return to the pre-treatment state within 1 to 2 months after the end of the administration [33]. Moreover, the use of perinatal antibiotics, such as in the intrapartum prophylaxis, influences the establishment of a normal gut microbial composition and function, in particular reducing the levels of bifidobacteria and increasing potential pathogens [34,35,36].

It is well established that a functional and balanced microbiota reflects a healthy condition of the host; on the other hand, an unhealthy status may be associated with a compromised gut microbiota displaying a decrease of beneficial bacteria and increase of harmful ones.

## 3. Probiotics with a Special Emphasis on *Bifidobacterium breve*

“Probiotic” means “for life” and it is currently used to name bacteria associated with beneficial effects for humans and animals. In 2001 the Food and Agriculture Organization of the United Nations (FAO)/World Health Organization (WHO) defined them as “live microorganism which, when administered in adequate amounts confer a health benefit on the host” [37]. This definition has been revised in 2014 by the International Scientific Association for Probiotics and Prebiotics, including in the term probiotic “microorganism for which there are scientific evidence of safety and efficacy” and excluding “live cultures associated with fermented foods for which there is no evidence of a health benefit” [38].

Probiotics that have been largely studied in humans include species of the *Lactobacillus* and *Bifidobacterium* genera. Probiotic administration in the first stage of life results to be more effective in prevention and treatment of disorders, leading to a correct microbial colonization when the gut microbiota is still in a period of establishment. Several studies have shown the beneficial effects of *Lactobacillus reuteri*, one of the most used probiotics in infants, for the prevention and treatment of infant gastrointestinal disorders, including colics, regurgitation, vomit, constipation [39,40,41]. This species has been demonstrated to improve symptoms and reduce the number of anaerobic Gram negative bacteria, *Enterobacteriaceae* and enterococci in colicky infants [42,43]. Furthermore, *L. reuteri* ATCC 55730 was effective in children with distal active ulcerative colitis (UC) improving mucosal inflammation and modulating mucosal expression levels of some cytokines involved in the bowel inflammation [44]. *Lactobacillus* and *Saccharomyces* strains (*L. casei* CG, *L. reuteri* ATCC 55730 and a strain of *S. boulardii*) exerted positive effects as supplement for rehydration therapy for infectious diarrhoea in children by reducing the duration and stool frequency [45].

Several data are available for the use of bifidobacteria as probiotics for therapeutic purposes in infants [46]. As an example, *Bifidobacterium* strains belonging to the *animalis* (BB-12 strain) and *longum* species proved their efficacy against acute rotavirus diarrhoea in hospitalized children, particularly by increasing the immune response and decreasing duration of disease [47,48,49]. In addition, administration of *Bifidobacterium bifidum* and *B. animalis* strains in preterm and low birth weight infants demonstrated clinical positive effects for treatment of necrotizing enterocolitis (NEC) [50,51,52].

Among the different species belonging to the *Bifidobacterium* genus, *Bifidobacterium breve,* is the dominant one in breast-fed newborns [53] and one of the most used in infants. The species *B. breve* was firstly described by Reuter [54], who isolated from breast-fed infant faeces and named seven species of *Bifidobacterium*, including *B. parvulorum* and *B. breve*. The two species were then combined under the name of *B. breve* [55]. *B. breve* strains are also found in the vagina of healthy women [54]. Their presence in extra-body environments is a consequence of faecal contamination and the species is a useful indicator of human and animal faecal pollution [56]. *B. breve*, like other *Bifidobacterium* species, possess an array of enzymes for the utilization of different carbohydrates. These enzymes, useful to adapt and compete in an environment with changing nutritional conditions, are inducible in the presence of specific substrates. Amongst them, glycosidases, neuraminidases, glucosidases, galactosidase are included as well as extracellular glycosidases that degrade intestinal mucin oligosaccharides and glycosphingolipids [57]. *B. breve* also possess a glucosidase with a β-D-fucosidase activity, useful for the utilization of fucosilated HMO [58]. *B. breve* is included in the list of Qualified Presumption of Safety (QPS) biological agents [59]. Furthermore, recent studies have shown that human milk, traditionally considered as sterile, contains commensal, mutualistic and/or potentially probiotic bacteria for the infant gut. Among the different *Bifidobacterium* species found in human milk, *B. breve* strains have been detected with DNA-based techniques and also isolated and characterized [60]. These bacteria from human milk rapidly colonize the newborn’s gut, protect the infant against infections and contribute to the maturation of the immune system [60].

Early studies by Akiyama et al. [61] showed that *B. breve* administration soon after birth was effective in developing a normal intestinal microbiota and, furthermore, *B. breve* showed a stronger affinity for immature bowel than other species, such as *B. longum*, evidencing its strong capabilities as probiotic. These achievements stimulated the development of further studies that gave new insights to the importance of this species as probiotic in infants.

Aloisio et al. [62] screened 46 *Bifidobacterium* strains for their capability of inhibiting the growth of gut pathogens including coliforms isolated from colicky infants. The most interesting strains belonged to the *B. breve* species, namely B632 strain (DSM 24706), B2274 strain (DSM 24707) and B7840 strain (DSM 24708). In addition to the antimicrobial activity against coliforms and other pathogenic bacteria, the strains did not possess transmissible antibiotic resistance traits and were not cytotoxic for gut epithelium, which are important pre-requisites for their use as probiotics. *B. breve* B632 was also able to stimulate the activity of mitochondrial dehydrogenases of macrophages and the production of IL-6, linked to a considerable activation of macrophages and endothelial cells in inflammatory condition. The potential of *B. breve* B632 as probiotic was also evidenced by Simone et al. [63]: it was able to inhibit the growth of *Enterobacteriaceae* in an *in vitro* gut model system stimulating the intestinal microbiota of a 2-month colicky infant, supporting the possibility to move to an *in vivo* study. Another strain of *B. breve*, BR03 (DSM 16604), revealed to be effective, as well as B632, in inhibiting the growth of 4 *E. coli* biotypes [64]. Mogna et al. [65] also underlined the validity of these two *B. breve* strains (B632 and BR03) in an *in vivo* study. The administration of both strains for 21 consecutive days as an oily suspension (daily dose of 100 million live cells of each strain) to healthy children was effective in obtaining gut colonization and in decreasing total faecal coliforms.

A biotechnological approach could improve the gastric transit survival, gastrointestinal persistence and therapeutic efficacy of the strain *B. breve* UCC2003, isolated from infant stool, via the heterologous expression of the listerial betaine uptake system gene, BetL [66]. In addition to the improved capability of colonizing the intestine of inoculated mice, the strain was also able to reduce *Listeria* proliferation in the organs of the infected mice. Although the introduction of genes from pathogens into probiotic cultures is unlikely to meet approval from regulatory authorities, this study underlined that probiotic characteristics can be susceptible to improvements. Future perspectives include the obtainment of BetL homologues from Generally Recognized as Safe (GRAS) organisms and natural selection of probiotic cultures with elevated expression of such homologues.

*B. breve* strain Yakult (BBG-01) is another widely used probiotic strain. It was one of the first *B. breve* strain shown to possess the ability to modulate the intestinal microbiota by reducing the count of several pathogenic bacteria, such as *Campylobacter*, *Candida* and *Enterococcus* spp., after oral administration [67,68]. This strain has also displayed an anti-infective activity against Shiga-toxin-producing *E. coli* (STEC) O157:H7 in infected mice [69].

For its valid properties as probiotic, *B. breve* has also found a notable place in food technology in the fermentation of milk. In this regard, the positive effects associated to *B. breve*-fermented soymilk has been reported in several studies, demonstrating to improve lipid metabolism, alcohol metabolism and mammary carcinogenesis in mice models [70,71,72].

Moreover, a strain of *B. breve* has been included in a widespread of commercial high concentrated probiotic preparation, known as VSL#3, which contains 10^11^–10^12^ viable lyophilized cells of different bacterial species that are usual component of human gut microbiota. Specifically, the formulation contains four strains of *Lactobacillus* (*L. paracasei*, *L. plantarum*. *L. acidophilus* and *L. delbrueckii* subsp. *bulgaricus*), three strains of *Bifidobacterium* (*B. longum*, *B. breve* and *B. infantis*) and one strain of *Streptococcus salivarius* subsp. *thermophilus*. VSL#3 exhibited an immunomodulatory capacity in *in vitro* studies by increasing the production of anti-inflammatory cytokines and inhibiting the production of pro-inflammatory cytokines [73].

## 4. *B. breve* Effectiveness in Mice Models

The strong evidence of the immune modulating capability of *B. breve* strains has been consolidated and well documented in a large number of animal models studies, which are the basis for human clinical trials.

The oral administration of *B. breve* YIT4064 strain, isolated from faeces of a healthy breast-fed infant, in mice immunized orally with an influenza virus was able to increase anti-influenza virus IgG levels in serum, thus protecting mice against infection. The authors concluded that the oral administration of this strain may enhance antigen-specific IgG against various pathogenic antigens taken orally and induce protection against various viral infections [74]. This conclusion was also supported by the study of Yasui et al. [75] that proved that the same strain stimulated anti-influenza virus hemagglutinin IgA by Peyer’s patch cells in response to addition of hemagglutinins. These antibodies may reach the mucosal tissue and prevent influenza virus infection.

*B. breve* UCC2003 possessed a cell surface exopolysaccharide (EPS) able to play an important role in immunomodulation in B cell response. Administration for 3 consecutive days of EPS^+^
*B. breve* strains in mice infected with *Citrobacter rodentium*, a diarrheagenic pathogen related to human *E. coli*, is effective in reducing the pathogen colonization, differently from mice fed with EPS^−^
*B. breve* [76]. EPS was involved in the production of a biofilm on the gut epithelium [77] preventing the attachment of *C. rodentium*.

Natividad et al. [78] illustrated the relationship between *B. breve* NCC2950 and regenerating (REG) III proteins, molecules belonging to the family of C-type lectins, which are expressed in the intestine and involved in maintaining gut homeostasis. The group REGIII-γ was measured in the ileum and colon of germ-free (GF) mice, mice colonized with specific pathogen free (SPF) microbiota and with a low diversity microbiota (altered Schaedler flora–ASF). Monocolonization with the probiotic *B. breve* NCC2950 but not with the commensal *E. coli* JM83, significantly induced REGIII-γ expression.

*B. breve* MRx0004, isolated from faeces of healthy humans, possessed a protective action in a severe asthma condition [79]. The study remarked an important decrease of neutrophil and eosinophil infiltration in lung bronchoalveolar lavage fluid in a mouse model of severe asthma after the probiotic treatment. This result, together with the demonstrated reduction of pro-inflammatory cytokines and chemokines involved in neutrophil migration, showed that *B. breve* MRx0004 effectiveness in reducing the above-mentioned inflammation condition paves the way for next-generation drug for management of severe asthma.

Many *B. breve* strains played an important role in prevention and treatment of various allergy conditions. Oral administration of *B. breve* M-16V, isolated from faecal sample of a healthy infant, in ovalbumin (OVA)-immunized mice significantly reduced the serum levels of total IgE, OVA-specific IgE and OVA-specific IgG1 and *ex vivo* production of IL-4 by the splenocytes [80]. Schouten et al. [81] showed that an intervention with a synbiotic formulation, comprising *B. breve* M-16V and a GOS/FOS mixture, was protective against the development of symptoms in mice orally sensitized with whey. The promising effect was confirmed by Kostadinova et al. [82] demonstrating the partially prevention of skin reaction due to cow’s milk allergy, following the probiotic administration in combination with specific β-lactoglobulin—derived peptides and a specific blend of short- and long-chain fructo-oligosaccharides in mice. Particularly, the treatment, besides increasing the cecal content of propionic and butyric acid, determined an increase of IL-22 expression, which plays an antimicrobial role in the innate immunity response and of the anti-inflammatory cytokine IL-10 in the Peyer’s patches. This outcome agrees with Jeon et al. [83], who demonstrated that the administration of the *B. breve* Yakult strain increased the number of IL-10-producing CD4^+^ T cells in the large intestine of murine models and an increased production of acetic acid [69].

*B. breve* was also involved in protective mechanisms against obesity; the orally administration of *B. breve* B-3 in a mouse model with diet-induced obesity could suppress the increase of body weight and epididymal fat, with improved serum levels of total cholesterol, fasting glucose and insulin and act by regulating gene expression pathways involved in lipid metabolism and response to stress in the liver [84,85].

Increasing evidence suggests that a brain–gut–microbiome axis exists, although its role in cognition remains relatively unexplored [6,86]. Bifidobacteria were found to improve the behavioural deficits and to possess a potential action on stress-related disorders in model mice [87]. *B. breve* strains potential has also been investigated for the capability of conferring beneficial effects on neurological diseases. Savignac et al. [88] showed that 6 weeks feeding of *B. breve* 1205 strain resulted in positive effects on compulsive behaviour in marble burying test, anxiolytic effects in the elevated plus maze and reduced body weight gain in model mice, contributing to a general amelioration of anxiety and metabolism. Kobayashi et al. [89] showed that oral administration of *B. breve* A1, isolated from faeces of human infants, prevented cognitive decline in Alzheimer disease (AD) model mice, with a reduction of neural inflammation; they observed that the probiotic provided ameliorations in both working memory and long-term memory. Furthermore, they found an increase of plasma acetate levels after the probiotic treatment and the neural inflammation reduction can be considered as a consequence of this increase due to *B. breve* administration, since SCFAs have been shown to have immune modulatory functions in model mice [90]. This evidence suggests that *B. breve* A1 has therapeutic potential for preventing cognitive impairment in Alzheimer disease and the necessity to move to a clinical intervention to evaluate the effects on diseased humans.

*B. breve* supplementation can affect the metabolism of fatty acids. Among them, eicosapentaenoic acid (EPA), which derives from α-linolenic acid metabolization, is an essential constituent of the cell membrane, plays an important role in brain and nervous system development and in inflammatory response [91]; docosahexanoic acid (DHA), which derives from EPA metabolization, is one of the major n-3 polyunsaturated fatty acids (PUFA) in the brain and is essential for a correct development of foetal encephalon [92]. Some studies revealed that human commensal microorganisms are able to synthetize bioactive isomers of conjugated linoleic acids (CLA) from free linoleic acid [93]; CLA was proven to possess antiatherosclerotic, antidiabetic and immunomodulatory properties [94,95]. Wall et al. [96] demonstrated that oral administration for 8 weeks to different animals (pigs and mice) of *B. breve* NCIMB 702258, a CLA producer strain, in combination with linoleic acid as substrate, increased the concentration of the predominant CLA isomer found in nature (*c*9, *t*11) in the liver. Furthermore, this supplementation in mice increased EPA and DHA levels in the adipose tissue and reduced proinflammatory cytokines tumour necrosis factor-α (TNF-α) and interferon-γ (IFN-γ) levels. The same authors demonstrated that a 8 weeks administration with the same *B. breve* strain and α-linolenic acid, the precursor of EPA, resulted in an increase in the liver EPA and brain DHA concentrations in mice. These results outline that the *B. breve* strain is a notable candidate for the treatment of inflammatory and neurodegenerative being able to modulate the hippocampal expression of brain-derived neutrophic factor (BDNF), a neurotrophin involved in development of the nervous system [97,98]. Particularly, the probiotic treatment reduced the expression of BDNF exon IV, which has been described as being highly responsive and increased by stress [99].

## 5. *B. breve* Application in Clinical Trials in Paediatrics

The use of *B. breve* strains for treatment and prevention of human diseases have been increasingly expanding in the last decade. Being bifidobacteria the most abundant bacterial group in infant gut, most of the studies are focused on paediatric subjects. Figure 1 summarizes the main applications of *B. breve* in paediatric diseases.

Therapeutic and protective role for human health of *B. breve* strains both as single strain or as a mixture of two strains of the same species has been demonstrated. As already mentioned, several researchers account for the improved efficacy of multi-species and multi-strain formulations that acting with a synergic effect, may enhance the effectiveness of each single strain [100,101].

### 5.1. Preterm Infants and Necrotising Enterocolitis (NEC)

A consistent number of preterm infants, especially those of very low birth weight, are subjected to episodes of systemic infection caused by antibiotic resistant bacteria and fungi that can lead to chronic diseases and brain injuries [102,103]. These episodes can result from a combination of factors, including immature gastrointestinal tract mucosal barrier and undeveloped gastrointestinal tract immune system, which may predispose premature infants to bacterial translocation, causing systemic infection and necrotising enterocolitis (NEC) [104,105]. In addition, preterm infants have revealed an altered microbiota composition, resulting in almost undetectable bifidobacteria counts during the first and second week of life, differently for those at term [106,107,108]. This observation has allowed the formulation of the hypothesis that a bifidobacteria treatment could lead to a reintegration of beneficial bacteria in the intestinal environment and a reduction of bacterial translocation to other districts, stimulating researches in this sector. One of the first study that investigated the effects of a *B. breve* supplementation in preterm neonates reported that the strain YIT4010, administered as a suspension of distilled water containing 0.5 × 10^9^ bacterial cells for 28 days, was able to colonize efficiently the intestinal tract, to reduce abnormal abdominal symptoms and to improve the weight gain [109]. A later study compared the effects of the administration of a *B. breve* strain a few hours after birth and 24 h after birth; the supplement was prepared by dissolving 1.6 × 10^8^ cells in 0.5 mL of 5% glucose solution and administered twice a day for all the duration of hospitalization [110]. In newborns administered with the probiotic soon after birth, bifidobacteria were detected significantly earlier and the number of *Enterobacteriaceae* at 2 weeks after birth was significantly lower, compared to the infants treated 24 h after birth demonstrating that a very early probiotic intervention may contribute to the establishment of a beneficial gut microbiota and the prevention of infectious diseases [110].

A more recent work proved the suitability of *B. breve* M-16V administration for routine use in preterm infants in order to control the gut microbiota colonization and shift it towards a healthy profile [111]. Moreover, a retrospective cohort study was performed with the purpose of evaluating whether the supplementation with the same probiotic to preterm neonates would reduce the risk of NEC [112]. NEC represents the most life-threatening pathology of preterm neonates with incidence and mortality of 10–12% and 40–45%, respectively. It is characterized by gastrointestinal dysfunction progressing to pneumatosis intestinalis, systemic shock and rapid death in severe cases [113,114]. NEC is categorized into 3 different stages based on the severity of the disease, from stage I, a suspicion for disease, to stage III, corresponding to a severe progression of the disease [115]. Although the pathogenesis of this condition remains obscure, some important prevention strategies have been adopted, such as the use of antenatal glucocorticoids, early preferential feeding with breast-milk, prevention and treatment of infections [116]. Since preterm infants have shown an intestinal reduction of total bifidobacteria and a predominance of facultative anaerobes, some of which potentially pathogens, until the 20th day of life, it has been suggested that a major etiological factor for NEC could be an altered microbiota composition [117]. Therefore, a probiotic treatment can be an additional strategy for NEC prevention. A 3-week *B. breve* M-16V supplementation (3 × 10^9^ CFU/day) has been associated with a lower incidence of NEC (≥stage II) in very low birth weight infants born before 34 weeks; the incidence in those born before 28 weeks resulted lower but not statistically significant [112]. Satoh et al. [118] had already demonstrated the efficacy of *B. breve* M-16V administration in preventing NEC in extremely low and very low birth weight infants: the probiotic was daily supplemented at a dose of 1 × 10^9^ CFU dissolved in breast milk or breast-mixed with formula milk several hours after birth and continued until discharge from hospital (achievement of body weight 2300 g or gestational age of 37 weeks); the treatment led to a significant reduction of infection and mortality rate.

Various studies suggested that an overproduction of SCFAs in the intestinal environment can lead to mucosal injuries, which may evolve in NEC in premature infants [119,120]. Wang et al. [121] demonstrated that a 4 weeks *B. breve* M-16V supplementation (1.6 × 10^8^ cells suspended in 0.5% glucose solution) was associated with a reduction of butyric acid levels in very and extremely low birth weight newborns. Since butyric acid increases the IL-8 secretion in enterocytes, condition that may lead to neutrophil invasion, a known hallmark of NEC, *B. breve* administration can be considered protective against NEC onset.

Immediately after delivery, some physiological changes, especially in the immunologic system, occur in newborns in order to adapt themselves to the new environment. *B. breve* M-16V, administered at 10^9^ cells in 0.5 mL of 5% glucose solution starting several hours after birth, can increase the transforming growth factor β1 (TGF-β1) signals in preterm infants [122]. This increase has a relevant importance as it is known to induce oral tolerance, exert anti-inflammatory effects, express mucosal IgA and promote epithelial cell proliferation and differentiation [123]. A further study investigated the preventive effects of the same *B. breve* strain against infections and sepsis in extremely and very low birth weight newborns. The probiotic consisted on a freeze-dried preparation with a dose of 10^9^ CFU dissolved in breast- or formula-milk; the development of infection and sepsis resulted significantly lower in the supplemented group compared with the non-supplemented one [124], highlighting once more the efficacy of a *B. breve* treatment in the prevention of developing infections, sepsis and NEC.

According to Braga et al. [125] the combined use of *B. breve* Yakult and *L. casei* was able to reduce the occurrence of NEC and was associated with an improvement in intestinal motility in newborns. The intervention started at the second day of life and continued for 30 days, provided *L. casei* and *B. breve* mixed to human milk in a daily dosage of 3.5 × 10^7^ and 3.5 × 10^9^ CFU, respectively. The number of NEC confirmed cases (≥stage II) was reduced upon probiotic treatment.

### 5.2. Gastrointestinal Disorders

A disorder that affects up to 30% of newborns in the first months of life is infant colic. It is characterized by paroxysmal, excessive and incontrollable crying without identifiable causes [126] representing a serious problem for the family and, in many cases, it can cause disorders later in life [127,128]. The aetiology remains obscure but an unbalanced intestinal microbiota has been suggested to play a role in the disease pathogenesis. Several studies support the use of probiotics as therapeutic or preventive agent against colics but very few clinical trials have been performed on bifidobacteria application. A mixture of *B. breve* strains (BR03 and B632), whose probiotic potential, as already highlighted in Section 3, has been extensively demonstrated *in vitro*, was prepared as oily suspension and administered at a daily dosage of 5 drops containing 10^8^ CFU of each strain to 83 infants, involving both breast and bottle-fed subjects [129]. Preliminary results showed that administration was effective in reducing minutes of daily crying. The clinical trial was then completed (155 infants, 130 breast- and 25 bottle-fed), as described in Aloisio et al. [130]; the *B. breve* mixture was able to prevent gastrointestinal disorders in healthy breast-fed infants, principally by reducing 56% of daily vomit frequency, decreasing 46.5% of daily evacuation over time and improving stool consistency. The strength of this study is the interrelation among a prolonged probiotic treatment, several clinical and anthropometric parameters (e.g., crying time, stool frequency, colour and consistency, regurgitation, vomits, weight, length, head circumference of newborn, delivery mode, type of feeding, gestational age) and main gut microbial groups. Epidemiological data have shown the predisposition of neonates born by caesarean section to develop obesity later in life [131,132]. However, the *B. breve* supplementation in infants born by caesarean section [130] resulted in a lower catch-up growth in weight, thus allowing the authors to speculate a protective effect of the probiotic strains against the risk to develop metabolic disturbance later in life.

Another common disease in childhood related to the intestinal tract is functional constipation, a chronic condition characterized by infrequent defecation (less than three times per week) and more than two episodes of faecal incontinence per week [133]; the pathogenesis, undoubtedly multifactorial, has not a well-defined aetiology. It has been shown that, despite intensive medical and behavioural therapy, 25% of patients developing constipation before the age of 5 years continue to have constipation upsets beyond puberty [134]. A pilot study showed the beneficial effects of 4 weeks treatment with *B. breve* Yakult (BBG-01) in constipated children: daily administration of 10^8^–10^9^ CFU led to a significantly increase in defecation frequency and amelioration of stool consistency, frequency of episodes of faecal incontinence and abdominal pain [135]. There is a debate of whether it is more effective the use of single strains or an association of them for constipation treatment; however, the mentioned study demonstrated that the intake of only one *B. breve* strain is even effective. Giannetti et al. [136] investigated the effects deriving from the administration of a mixture of 3 bifidobacteria, namely *B. infantis* M-63, *B. breve* M-16V and *B. longum* BB536, in children suffering from irritable bowel syndrome (IBS). IBS is a functional bowel disorder characterized by chronic abdominal pain, discomfort, bloating and altered bowel habits including diarrhoea or constipation [137]. The daily dose was about 10^9^ cells for each strain administered as bacterial powder and the treatment lasted 6 weeks. The bifidobacteria mixture intake resulted in a significant decrease in prevalence and frequency of abdominal pain and an improvement of the quality of life, assessed by an interview-administered validated questionnaire.

The commercial formulation VSL#3, already described in Section 3, was used in several clinical studies targeted to different diseases in paediatrics resulting in an amelioration of the health status of children suffering from IBS [138]. In this randomized, double-blind, placebo-controlled, multicentre trial, patients were treated with one sachet (twice in those 12–18 years old) of probiotic mixture containing 4.5 × 10^12^ bacteria for 6 weeks. The preparation was effective in improving the overall perception of symptoms, the severity and frequency of abdominal pain, abdominal bloating and family assessment of life disruption, leading to a general improving of quality of life in children suffering from IBS.

Miele et al. [139] carried out the first paediatric, randomized, placebo-controlled trial using VSL#3 for the treating of ulcerative colitis (UC). This disorder belongs to the chronic inflammatory bowel disease (IBD) category, has a prevalence of about 100 cases per 100,000 children [140] and occurs as diffuse mucosal inflammation in the colon; it is characterized by periods of remission and relapse episodes, not all the patients tolerate the existing treatment to induce remission for their adverse effects and in 20–30% of paediatric patients failure of the treatment occurs [141]. Since the pathogenesis, beside genetic susceptibility, is linked to compromised immune response and alteration in gut microbiota composition, the idea beyond the study was that 1 year of VSL#3 administration might improve the health status of patients. Subjects with an average age of 10 were supplemented with a weight-base dose of probiotic (4.5 × 10^11^–1.8 × 10^12^ bacteria per day); treated patients showed a significantly higher rate of remission compared to placebo and a significantly lower incidence of relapse within 1 year of follow-up. According to the authors, this success may be related to the use of a mixture of various probiotics, which might have a strong synergic action and to the high bacterial concentration of viable cells contained in the mixture. Furthermore, the probiotic preparation showed to be safe and well tolerable by children with a diagnosis of UC.

The efficacy of VSL#3 in paediatric diseases was also evaluated by Dubey et al. [142], who conducted a double-blind, randomized, placebo-controlled trial treating acute rotavirus diarrhoea in children. VSL#3, containing a total of 9 × 10^9^ bacteria/dose and administered for 4 days, significantly reduced, already on day 2, mean stool frequency and improved stool consistency; these results were also reflected in the lower volume of oral rehydration salts administered in children who received the probiotic. The functional role of VSL#3 was investigated by Sinha et al. [143], who focused on the prevention of neonatal sepsis in low birth weight infants, one of the infections which evolves more rapidly in this paediatric category. The mixture, containing 10^9^ bacteria/dose, was administered for 30 days. VSL#3 intake in low birth weight was associated with a non-significant 21% reduction in the risk of suspected sepsis; nevertheless, in the sub-group of infants weighing 1.5–1.99 kg, the reduction of the risk of suspected sepsis was statistically significant, differently from newborns weighing 2.0–2.49 kg. The results of the study allowed to conclude that the intervention may be useful for the most vulnerable subjects of low birth weight.

As infant feeding has a crucial role in developing infant gut microbiota and consequently intestinal immunity, fermented formula milk containing probiotics or prebiotics has been developed. This approach is aimed at protecting infants from various gastrointestinal disorders by modulating gut microbial composition. The first study that evaluated the effects of a fermented formula milk with *B. breve* C50 and *Streptococcus thermophilus* 065 on the incidence of acute diarrhoea in healthy infants was a randomized, double-blind, placebo-controlled multicentre study, which involved 971 subjects belonging to three different areas of France [144]. The trial was planned to occur in a high risk predicted period for diarrhoea incidence in France (from October to January) and the supplementation lasted 5 months. Although no reduction in the incidence and duration of diarrhoea episodes were observed after the intervention, a lower number of dehydration cases, a lower number of medical consultation cases with fewer oral rehydration solution prescriptions and changes of formula were registered. These outcomes can be considered as indicators of probiotic positive effects on the severity of the disease. According to the authors, these results may be related to the bifidogenic and immunomodulatory properties of fermentation products contained in formula-milk.

### 5.3. Celiac Disease

The efficacy of the probiotic mixture containing *B. breve* B632 and *B. breve* BR03 was also shown in children affected by celiac disease. In this case, the strains were administered as lyophilized powder at a daily dosage of 10^9^ CFU of each strain for 3 months in celiac children on a gluten free diet (GFD). A preliminary important outcome obtained from the intervention was the reduction of pro-inflammatory cytokine TNF-α in blood samples of celiac children on GFD [145]. The gut microbiota composition was also studied with Next Generation Sequencing (NGS) technology. Unexpectedly, the intervention did not cause changes at the level of the genus or phylum to which the administered probiotics belong but the probiotic acted as a “trigger” element for the increase of *Firmicutes* and the restoration of the physiological *Firmicutes*/*Bacteroides* ratio that was altered in celiacs with respect to healthy subjects. Moreover, the intervention restored the normal amount of *Lactobacillaceae* members, reaching almost the same values of healthy subjects [146]. Besides modulating inflammatory condition and gut microbiota composition of celiac children, *B. breve* supplementation influenced the SCFAs profile; acetic acid had a negative correlation with *Verrucomicrobia*, *Euryarcheota* and particularly *Synergisestes* [147]. Although *Synergisestes* is a minor phylum in human faeces (abundance of 0.01%) of healthy subjects, it was found to have a considerable role for human health because of its negative correlation with TNF-α that may indicate an anti-inflammatory role [148,149]. In the study of Primec et al. [147], the *Synergisestes* phylum clearly confirmed its anti-inflammatory role negatively correlating with pro-inflammatory acetic acid after three months of probiotic treatment.

### 5.4. Paediatric Obesity

Another pathology in which the gut microbiota may play a notable role is obesity. Although it is accepted that obesity results from disequilibrium between energy intake and expenditure, it is a complex disease and not completely understood. Nowadays, obesity prevalence is spreading especially among children and adolescents and it can be considered a worldwide epidemic. Obesity has been associated with a chronic inflammation that may conduct to insulin resistance [150,151]. Recently, obesity has been associated with a specific profile of the gut microbiota characterized by lower levels of bacteria belonging to *Bacteroides* and *Bifidobacterium* genera compared to that of lean individuals [152]. In addition, bifidobacteria were shown to be higher in children maintaining normal weight at 7 years old than in children developing overweight and their administration was able to reduce serum and liver triglyceride levels and to decrease hepatic adiposity [153,154]. The mixture of *B. breve* already mentioned (BR03 and B632) was used in a cross-over double-blind randomized controlled trial in order to re-establish metabolic homeostasis and reduce chronic inflammation in obese children [155]. Although the study is still on-going, preliminary results related to the part previous the cross-over demonstrated that a *B. breve* administration in obese children is promising: 8 weeks treatment seems to ameliorate glucose metabolism and could help in weight management by reducing BMI, waist to height ratio and waist circumference [155].

### 5.5. Allergies

There are increasing evidences that the intestinal microbiota plays an important role in the development of allergic diseases, in particular, low bifidobacteria levels appear to be associated with atopic dermatitis [156]; in the previous section, the potential of *B. breve* in preventing and treating allergy conditions was reported and this impressive role has been confirmed in clinical studies. *B. breve* M-16V revealed to be effective in the treatment of cow’s milk hypersensitivity infants with atopic dermatitis [157]. *B. breve*, added to the casein-hydrolysed milk formula at the dosage of 5 × 10^9^ CFU or 15 × 10^9^ CFU per day, increased the proportion of bifidobacteria in the gut microbial composition and ameliorated allergic symptoms by interacting with the immune system and no remarkable dose dependent differences were detected [157]. The synergetic combination of probiotics and prebiotics, known as synbiotic, seems also to be promising in atopic dermatitis treatment. In this regard, Van der Aa et al. [158] studied the effects of a synbiotic mixture on atopic dermatitis in formula-fed infants; the formulation consisted of *B. breve* M-16V at a dose of 1.3 × 10^9^ CFU/100 mL and a mixture of 90% short-chain galactooligosaccharides (scGOS) and 10% long-chain fructooligosaccharides (IcFOS), 0.8 g/100 mL added to formula milk. Although the formulation, administered for 12 weeks, had no effect on atopic dermatitis severity, it significantly modulated the composition and the metabolic activity of gut microbiota, leading to a decrease of pH, high lactate and low butyric levels resembling the metabolic profile of breast-fed infants [159]. The same synbiotic mixture has demonstrated to reduce the prevalence of asthma-like symptoms and the prevalence of asthma medications use after the fulfilment of a 1-year follow-up [160].

The effects of a formulation containing *B. breve* M-16V and *B. longum* BB536 for the prevention of allergies in infants enrolling both mothers and newborns was studied [161]. The formulation was provided as powder daily doses containing 5 × 10^9^ CFU/g of each strain. Pregnant women begun the supplementation 4 weeks before the expected date of delivery and the newborns received the probiotic mixed to water, breast- or formula-milk starting 1 week after birth and continuing for 6 months. The study revealed that prenatal and postnatal supplementation with a bifidobacteria mixture reduced the risk of developing eczema and atopic dermatitis in infants. NGS analyses of newborns’ faecal samples showed significant differences of the major intestinal microbial phyla (*Actinobacteria*, *Bacteroidetes*, *Proteobacteria*) of allergic and non-allergic infants at 4 months of age. However, these differences were lost at 10 months of age, highlighting that the microbiota of early stages is particularly important in regulating allergies upset in infants.

### 5.6. Surgical Procedures

Surgical procedures can also alter gut microbiota composition and functions and disrupt intestinal barrier function, inducing the patient in a condition at risk for infection [162]. A probiotic therapy may be functional for patients improving the immunological function of the intestine and competing against harmful bacteria infection. A pilot study demonstrated that daily administration of *B. breve* Yakult BBG-01 (10^9^ freeze-dried cells per day) to children younger than 15 years 7 days before surgery until discharge from hospital, simultaneously to intravenous antibiotics postoperatively treatment, reduced the incidence of bacteria in blood samples. Moreover, the intestinal microbial composition was improved by increasing *Bifidobacterium* spp. and reducing potential pathogens such as *Clostridium difficile*, *Pseudomonas* and *Enterobacteriaceae*. Higher concentrations of faecal acetate and lower faecal pH levels were detected in children who received the probiotic 2 weeks after surgery [163]. Improvement of intestinal environment resulting from a perioperative supplementation with the same strain was also observed in neonates undergoing surgery for congenital heart disease [164]. Daily dosage of 3 × 10^9^ CFU of *B. breve* Yakult (BBG-01) was administered starting 1 week before surgery and ending 1 week after the operation; infants who received the probiotic supplement showed significantly higher bifidobacteria levels and lower *Enterobacteriaceae*, *Staphylococcus* and *Pseudomonas* levels in faecal microbiota compared to infants not receiving the supplement. Moreover, probiotic treated infants exhibited significantly higher concentration of total organic acids levels compared to non-treated ones, in particular acetic acid increased immediately and 1 week after surgery; furthermore, the faecal pH tended to decrease with the probiotic intervention.

Kanamori et al. [165] documented in a case-report the efficacy of a synbiotic therapy, consisting in a combination of *B. breve* Yakult (BBG-01), *L. casei* Shirota and galactoolicosaccharides as prebiotic components, in a newborn with short bowel syndrome resulting from a consistent bowel resection performed soon after delivery. Patients affected from this pathology are subjected to an intestinal bacteria overgrowth due to their dilated intestine [166]; this condition can lead to a bacteria translocation in other districts inducing catheter sepsis, compromised carbohydrates fermentation resulting in high level of lactate, with consequent acidosis [167] and a possible incontrollable growth of intestinal pathogens. One year of synbiotic therapy, consisting in 3 g of bacteria (1 × 10^9^ bacteria/g per each strain) and 3 g of prebiotic per day, improved the nutritional state, prior compromised, by increasing the intestinal motility and suppressed the intestinal pathogen overgrowth, in particular *E. coli* and *Candida* spp.

The same synbiotic combination was used as a therapy for refractory and repetitive enterocolitis [168]; this disorder often occurs in paediatric surgery patients and the severe type may be fatal. The 7 recruited patients, having short bowels as a result of surgical resection and suffering from repetitive enterocolitis, were administered with 1 g of probiotic (10^9^ bacteria/g) 3 times daily for 36 months. All patients had an altered gut microbial composition prior to the therapy characterized by low levels of anaerobic bacteria and high levels of resident pathogenic bacteria. In spite of the frequent antibiotic treatments to which patients were exposed, the long synbiotic administration was effective in highly increasing bifdobacteria and lactobacilli levels, which were almost undetectable before the supplementation and incrementing faecal SCFAs, inducing a more normal ecosystem profile in the intestine. Moreover, most of patients accelerated their body weight gain and showed increased serum rapid turnover, with a general amelioration of their health status.

With the developing of therapies and surgeries in the field of perinatal and foetal cares, neonate survival outcomes have extraordinary increased; newborns that are subjected to these interventions need prolonged intensive care periods, which include use of antibiotics, respiratory care and restriction of enteral feeding. All these factors may affect the normal microbial gut colonization leading to severe infection and malnutrition [169]. A synbiotic therapy, including *B. breve*, as already observed, could be effective in preventing or correcting an abnormal microbial colonization in intensive care newborns. The same synbiotic therapy, largely and positively tested, including *B. breve* Yakult, *L. casei* Shirota and galactooligosaccharides, was applied to newborns with diagnosis of severe congenital anomalies [169]. The product contained 10^9^–10^10^ bacteria/g and was administered immediately after birth via a nasogastric tube, as soon as intestinal feeding was possible, first at a dose of 0.12 g per day in four equal dose and then, when the amount of milk increased, at 3 g per day in three equal doses. As results of the therapy, none of patients manifested enterocolitis, they showed an improvement in their clinical course and reached a body weight gain equivalent to that of normal infants. This last outcome has been hypothesized to be linked to the potential metabolic activity of the administered probiotics to promote liver lipogenesis and fat storage in the peripheral fat tissue contributing to the growth observed in these infants despite the congenital disorders [170].

### 5.7. Coadjuvant in Chemotherapic Treatment

A condition in which the use of probiotics may have a reliving effect is chemotherapy. The cancer itself and the drug-therapy inducing bone marrow suppression lead to an immunocompromising state in which an infectious could be fatal. Since the main source of infection is endogenous intestinal harmful bacteria [171], a probiotic treatment can certainly benefit the patient’s state by not only competing against pathogens for nutrients and attachments sites but also by stimulating gut immunity, producing organic acids and improving transepithelial resistance [172]. A study conducted in 2009 evaluated the effect of *B. breve* Yakult (BBG-01) strain in cancer paediatric subjects, administered with 10^9^ freeze-dried cells, corn starch and hydroxipropyl cellulose in 1 g of formulation. The administration was found to be effective in reducing febrile episodes, which may be the only sign of infection and the use of intravenous antibiotics by stabilizing the intestinal microbial composition [173].

An overview in chronological order of *B. breve* applications as a single strain and as a component of a multi-strain/multi-species formulation is reported in Table 1 and Table 2, respectively.

## 6. *B. breve* Administration in Adults: A Short Outcome

The use of *B. breve* has been largely investigated in paediatric scenery and its therapeutic role has been strongly supported by significant and solid outcomes; its use is not limited to paediatric supplementation but it is also involved in improving health condition in briefly outlined.

Minami et al. [174] investigated the use of *B. breve* B-3 at a daily dosage of 5 × 10^10^ CFU/capsule for 12 weeks in adults with a tendency for obesity. A significant decrease of the fat mass and an amelioration of blood parameters were observed, in particular a significant reduction of γ-glutamyltranspeptidase (γ-GTP), a marker used to evaluate liver injury and high-sensitivity protein C-reactive (hCRP), a marker used to evaluate the inflammatory reaction, were detected. Interestingly, a significant negative correlation between the value of fat mass and 1,5-anhydroglucitol, a marker that closely reflect short-term glucose status and glycaemic variability, was recorded suggesting the potential role of *B. breve* in the improvement of diabetes.

Ishikawa et al. [175] showed the effects of one year of *B. breve* Yakult treatment, in association with galactooligosaccharides as prebiotic, in patients diagnosed with UC. The probiotic, containing 10^9^ CFU/dose of freeze-dried powder, was administered immediately after every meal 3 times a day and the prebiotic, at a dosage of 5.5 g, was administered once a day. The synbiotic intervention improved the endoscopic score by decreasing the values of severity mucosa damage [176] and reduced the level of myeloperoxidase, which is secreted by neutrophils and macrophages accumulated in the inflamed lesions and positively correlated with the disease severity [177]. Regarding gut environment results, the synbiotic treatment significantly reduced *Bacteroidaceae* counts and faecal pH, which may be connected to an increment of faecal SCFAs.

An interesting relationship was evaluated by Kano et al. [178]: since a Japanese 2007 survey evidenced that women who suffer from abnormal bowel movements also showed skin disorders, they conducted a double-blind, placebo-controlled, randomized trial to investigate the effects of probiotic and prebiotic fermented milk on skin of healthy adult women. The fermented milk contained galactooligosaccharides, polydextrose, *B. breve* Yakult, *Lactococcus lactis* and *S. thermophilus* at a daily dose of 6 × 10^10^, 5 × 10^10^, 5 × 10^10^ CFU/100 mL of milk, respectively. The synbiotic intake, which lasted 4 weeks, resulted to prevent hydration level decreases in the stratum corneum. The intervention increased cathepsin L-like protease activity, which can be considered as an indicator of keratocyte differentiation, as proteolysis of cathepsin L activates transglutaminase 3, which plays an important role in the *stratum corneum* formation [179]. Moreover, the administration reduced phenol levels in serum and urine and since the production of phenols is inhibited at low intestinal pH, an increase of intestinal organic acid levels might be occurred after the treatment.

The probiotic preparation VSL#3 has been extensively used for the treatment of IBD in adulthood. Brigidi et al. [180] investigated the effects of 20 days VSL#3 administration in patients with diarrhoea predominant-IBS or functional diarrhoea; the probiotic intake caused changes in gut microbiota composition with a significantly increase of total lactobacilli, total bifidobacteria and *S. thermophilus*, which are component of VSL#3. The treatment led also to an improvement of some enzymes functions, whose actions are compromised in IBD, by reducing urease activity, whose products usually allow pathogenic bacteria to survive in the gastrointestinal tract and contribute to mucosal tissue damages [181] and by increasing β-galactosidase activity, which is involved in the metabolism of unabsorbed carbohydrates. Pronio et al. [182] confirmed the positive role of VSL#3 upon treatment of patients undergoing ileal pouch anal anastomosis for ulcerative colitis. The probiotic intervention reduced signs and symptoms of inflammation inducing a significant expansion of cells associated to an improvement of the inflammatory condition of the pouch mucosa. An interesting microbial outcome was evidenced by Kühbacher et al. [183]: the UC remission maintained by VSL#3 administration was accompanied by a higher bacterial diversity actually not related to the probiotic intake. However, the increase of bacterial diversity may represent a therapeutic mechanism that supports the VSL#3 activity in maintaining UC remission. Bibiloni et al. [184] showed that 6 weeks administration with the probiotic mixture improved UC remission and response in patients not responding to traditional therapy. Since VSL#3 has been demonstrated to maintain remission in UC patients intolerant or allergic to 5-aminosalicylic acid (5-ASA), known also as mesalazine [185], Tursi et al. [186] demonstrated the efficacy on UC of another therapeutic combination: VSL#3, in association with balsalazide, 5-ASA prodrug, was shown to be significantly superior to balsalazide alone and to mesalazide in the treatment of active mild-to-moderate UC. One of the key points of the study is the low dosage of balsalazide used (2.25 g/day), usually not effective in reducing UC symptoms and inducing remission. Therefore, the low dosage appeared to be effective only in combination with VSL#3. In this regard, a more recent study, involving a larger number of patients, highlighted the superior ability of VSL#3 to improve relapsing mild-to-moderate UC when added to standard UC treatment with respect to patients on standard treatment only, confirming the potential synergic action exerted by standard UC pharmacological treatments and VSL#3 [187]. The reason for this synergic action may be a combined effect of the chemotherapic on the disease and of the probiotic on the general well-being of the host. Clinical studies proved that this probiotic mixture was particularly effective in the treatment of IBD, improving abdominal pain duration and distention severity score in patients suffering from IBS [188]. Moreover, it was effective in clinical condition of diarrhoea-predominant IBS subjects [189,190].

## 7. Conclusions

This review has outlined the large number of cases in which *B. breve* strains, mainly as single strains but also in combination with other *Bifidobacterium* species or *Lactobacillus* strains, are used for therapeutic and prevention purposes and/or to prevent further complications of the disease in the paediatric sector. The analysis of the outlined results allows to conclude that, whereas *in vitro* or animal-model study are performed with a large number of different *B. breve* strains, clinical studies are performed with a restricted number of strains (mainly *B. breve* YIT4010, M-16V, the associations B632/BR03 and Yakult BBG-01). Therefore, there is the opportunity of expanding the potentialities of the strains used in clinical studies on the basis of the positive results obtained in pre-clinical studies and, therefore, more opportunities for a further development of “therapeutic microbiology.” A second interesting aspect outlined in this review is the frequent association of the *B. breve* administration with traditional chemotherapeutic treatment. This is particularly important in the treatment of very serious diseases in which stopping the traditional therapies may be considered risky for the patient. The probiotic can act as a supplement to prevent complication and improve the general health status of the patient. We are all confident that the improvement in the “therapeutic microbiology” sector will be a great aid to medical approach in the near future.

## Figures and Tables

**Figure 1 nutrients-10-01723-f001:**
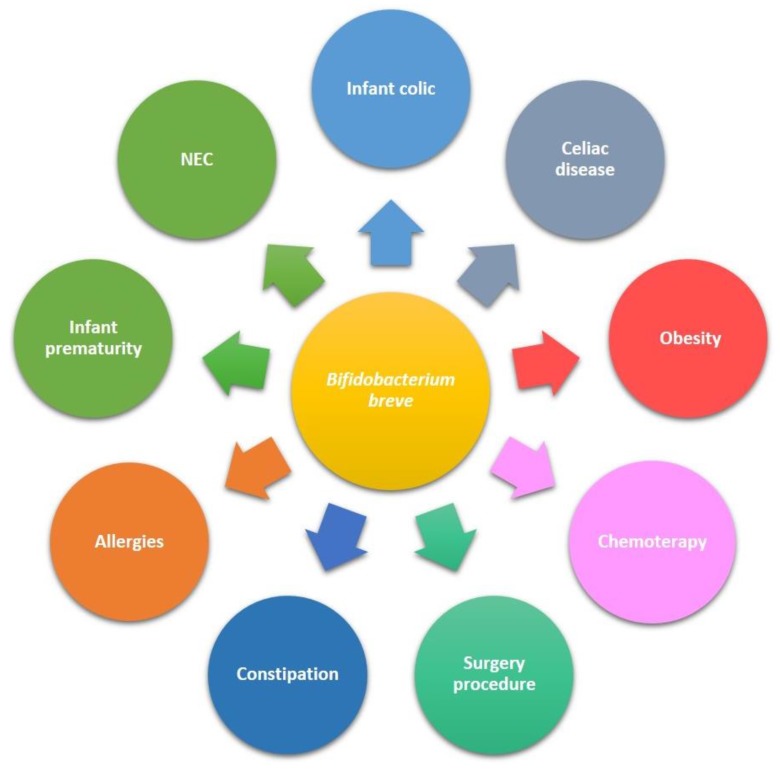
Paediatric diseases in which an amelioration of symptoms has been obtained upon *B. breve* strains administration.

**Table 1 nutrients-10-01723-t001:** Overview of *B. breve* strains applications in *in vitro* studies, mice model and paediatric trials.

*B. breve* Strains	Reported Effect(s)	References
*B. breve* B632	Strong antimicrobial activity against pathogens, stimulation of mitochondrial dehydrogenase activity of macrophages, stimulation of proinflammatory cytokines production in *in vitro* study	[62]
*B. breve* BR03	Inhibition of the growth of 4 *E. coli* biotypes in *in vitro* study	[64]
*B. breve* B632 + *B. breve* BR03	Reduction of total faecal coliforms in healthy children	[65]
	Reduction of pro-inflammatory TNF-α in blood samples of celiac children	[145]
	Reduction of minutes of daily crying in healthy infants	[129]
	Restoration of the healthy percentage of main gut microbial components in celiac children	[146]
	Improvement of glucose metabolism and weight management in obese children	[155]
	Reduction of daily vomit frequency, daily evacuation, improved stool consistency, protection against developing metabolic disturbance in healthy infants	[130]
	Modulation of faecal SCFAs profile in celiac children	[147]
*B. breve* Yakult (BBG-01)	Anti-infective activity against Shiga-toxin-producing *E. coli* in mice model	[69]
	Reduction of febrile episodes and use of intravenous antibiotics in cancer paediatric subjects	[173]
	Improvement of composition and metabolic activity of gut microbiota and reduction of incidence of bacteria in blood in paediatric surgery subjects	[163]
	Increased defecation frequency, improvement of stool consistency, frequency episodes of faecal incontinence and abdominal pain in constipated children	[133]
	Stimulation of anti-inflammatory IL-10-producing CD4+T cells in mice model	[83]
	Improvement of composition and metabolic activity of gut microbiota in paediatric surgery infants with congenital heart disease	[164]
*B. breve* YIT4064	Stimulation of anti-influenza virus hemagglutinin IgA production by Peyer’s patch cells in mice model	[75]
	Stimulation of antigen-specific IgG production against pathogenic antigens in mice model	[74]
*B. breve* UCC2003	Reduction of *Citrobacter rodentium* gut colonization in mice model	[76]
*B. breve* NCC2950	Induction of REGIII-γ expression in mice model and REGIII-α in *in vitro* study	[78]
*B. breve* MRx0004	Reduction of pro-inflammatory cytokines and lung neutrophil and eosinophil infiltration in severe asthma mice model	[79]
*B. breve* M-16V	Improvement of allergic symptoms associated to cow’s milk hypersensitivity in infants	[157]
	Immunomodulation activity by increasing TGF-β1 in preterm infants	[122]
	Reduction of infections and mortality for NEC in extremely and very low birth weight infants	[118]
	Reduction of faecal butyric acid in extremely and very low birth weight infants	[121]
	Reduction of total IgE, OVA-specific IgE and OVA-specific IgG in mice model	[80]
	Protection against developing of whey allergy symptoms in model mice	[81]
	Reduction of infections and sepsis incidence in extremely and very low birth weight infants	[124]
	Improvement of composition and metabolic activity of gut microbiota in infants with atopic dermatitis	[158]
	Reduction of asthma-like symptoms prevalence and asthma medication use prevalence in infants with atopic dermatitis	[160]
	Shifted gut microbiota towards a healthy profile in preterm infants	[111]
	Low incidence of NEC (≥stage II) in very low birth weight infants	[112]
	Partially protection against developing skin reaction due to cow’s milk allergy, increased cecal content of butyrate and propionate and increased antimicrobial IL-22 expression in mice model	[82]
*B. breve* B-3	Suppression of epididymal fat and body weight gain in mice model with diet-induced obesity	[84,85]
*B. breve* 1205	Amelioration of anxiety condition and general metabolism in mice model	[88]
*B. breve* A1	Prevention of cognitive decline in Alzheimer disease and reduction of neural inflammation in mice model	[89]
*B. breve* NCIMB 702258	Increased CLA isomer (*c*9, *t*11), EPA and DHA in adipose tissue and reduced proinflammatory cytokines in mice model	[96]
*B. breve* YIT4010	Reduced abdominal symptoms and improved weight gain in preterm infants	[109]
	Establishment of beneficial gut microbiota and prevention of infections in preterm infant	[110]

**Table 2 nutrients-10-01723-t002:** Overview of applications of *B. breve* strains combined to other bacterial strains in paediatric trials.

*B. breve* Strains	Probiotic Mixture	Reported Effect(s)	References
*B. breve* M-16V	*B. breve* M-16V *B. longum* BB536	Reduction of developing eczema and atopic dermatitis in infants	[161]
	*B. breve* M-16V *B. infantis* M-63 *B. longum* BB536	Reduction of abdominal pain prevalence and frequency, improvement of quality of life in IBS children	[134]
*B. breve* Yakult (BBG-01)	*B. breve* Yakult *L. casei* Shirota	Improvement of composition and metabolic activity of gut microbiota and of overall health status in infants with short bowel syndrome	[165,168]
		Prevention of enterocolitis, improvement of body weight and clinical course in infants with congenital disorders	[169]
	*B. breve* Yakult *L. casei*	Reduction of NEC incidence and improvement of intestinal motility in infants	[125]
*B. breve* C50	*B. breve* C50 *S. thermophilus* 065	Reduction of number of dehydration cases and medical consultation cases in children exposed to risk of developing acute diarrhoea	[144]
*B. breve* DSM 24732	VSL#3	Reduction of stool frequency and improving of stool consistency in children with acute rotavirus diarrhoea	[142]
		Manifestation of high rate of remission and low incidence of relapse in UC children	[139]
		Improvement of symptoms, severity and frequency of abdominal pain and bloating and family assessment of life disruption in IBS children	[138]
		Reduction of the risk of suspected sepsis in most vulnerable very low birth weight infants	[144]

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
