# Peer review of "Therapeutic Microbiology: The Role of Bifidobacterium breve as Food Supplement for the Prevention/Treatment of Paediatric Diseases"

_nutrients, 2018, doi:10.3390/nu10111723_

Reviewer 1 Report

The manuscript is very well written.

The authors should summarise what the paper adds to the existing knowledge. 

First seven pages are not strictly connected with “paediatric” as emphasised in the title.

Authors could enrich the text with scheme presenting the action of B. breve in paediatrics.

Detailed comments:

Lines 135-138:

There is more recent definition of ‘probiotics’ and it should be cited:

Hill, C., Guarner, F., Reid, G., Gibson, G. R., Marenstein, D. J., Pot, B., Morelli, L., Canani, R. B., Flint, H. J., Salminen, S., Calder, P. C., Sanders, M. E. (2014). The International Scientific Association for Probiotics and Prebiotics consensus statement on the scope and appropriate use of the term probiotic. Nature Rev. Gastroenterol. Hepatol. 11:506-514

Line 99:

There is “Staphilococcus” and should be “Staphylococcus”.

Lines 159-161; 163-168:

What strains exactly?

Line 226:

“GRAS” should be developed as “Generally Recognized as Safe”.

Lines 738-741:

What is the reason?

Author Response

Response to Reviewer 1 Comments

REVIEWER 1) The manuscript is very well written.

The authors should summarise what the paper adds to the existing knowledge. 

Response: A short summary of the novelty of the review is reported in lines 45-48

First seven pages are not strictly connected with “paediatric” as emphasised in the title.

Response: We reduced the paragraphs 1, 2, 3 and 4 in order to mainly emphasize the use of B. breve in paediatrics (overall, about 1 page reduction)

Authors could enrich the text with scheme presenting the action of B. breve in paediatrics.

Response: We added a scheme (Figure 1) that showes the main applications of B. breve in pediatric diseases.

Detailed comments:

Lines 135-138:

There is more recent definition of ‘probiotics’ and it should be cited:

Hill, C., Guarner, F., Reid, G., Gibson, G. R., Marenstein, D. J., Pot, B., Morelli, L., Canani, R. B., Flint, H. J., Salminen, S., Calder, P. C., Sanders, M. E. (2014). The International Scientific Association for Probiotics and Prebiotics consensus statement on the scope and appropriate use of the term probiotic. Nature Rev. Gastroenterol. Hepatol. 11:506-514

Response: Thank you for the suggestion, we provided for the insertion of the new aspects of “probiotics” definition and the related citation has been inserted.  

 Line 99:

There is “Staphilococcus” and should be “Staphylococcus”.

Response: We provided for the correction

 Lines 159-161; 163-168:

What strains exactly?

Response: for the first study that we reported (Oliva et al. 2012), focused on the treatment of active ulcerative colitis (UC) in children, the administered probiotic strain was Lactobacillus reuteri ATCC 55730. The strain has been added (lines 140-141).

For the second cited study (Allen et al. 2010), focused on the treatment of acute infectious diarrhea in children, the probiotics used were Lactobacillus casei strain GG, Lactobacillus reuteri strain ATCC 55730 and a non-specified strain of Saccharomyces boulardii. The strains (where available) have been added (lines 143-144).

Regarding effective Bifidobacteria against acute rotavirus diarrhea in hospitalized children, we added the strains where available. For some of the bifidobacteria used the strains were not specified in the original papers.

 Line 226:

“GRAS” should be developed as “Generally Recognized as Safe”.

Response: We provided for the correction

 Lines 738-741:

What is the reason?

Response: A possible explanation has been added in the text: “The reason for this synergic action may be a combined effect of the chemoterapic on the disease and of the probiotic on the general well-being of the host” (lines 703-704).

Reviewer 2 Report

Overall, the review is very thorough and it is clear that the authors put an immense amount of work into compiling this.  However, this also makes the review very dense.  A few paragraphs which do not touch on pediatric issues could be removed or reduced.  It is also very difficult to glean a thorough understanding of the potential for Bifidobacterium pertaining to different body systems or disease etiologies, as the information on any one area (ex. GI diseases) treated by probiotics is scattered across several sections.  I would suggest reorganizing sections (starting from line 162) into disease type and applications of Bifidobacterium, ex. “gastrointestinal disorders”, “nutritional status and development”, “respiratory infections”, “allergies and immune system development”.  This would aid the reader in appreciating the amount of work that has been done on Bifidobacterium as a probiotic, and dramatically improve the readability of the manuscript which is currently disjointed.  While this will require a major revision of the text layout, it will not require a major revision of the scope or content of the manuscript. 

Line 12: correct to “for human health”

Line 13: “gut microbiota”?

Line 14: “not apparently liked to the gut environment” is a bit vague, as I assume the authors mean “not linked directly to host biology”?

Line 37: remove comma after species

Line 55: I don’t think the 15,000 – 36,000 species estimate is accurate, and at best the original citation is probably pointing to sequence variants or strains.  That calculation is based on two clone-library studies and one early sequencing study, the highest of which estimated about 4,000 bacterial phylotypes, and all three studies use a relatively small library from which to estimate rarefaction curves.  Given the effect of sampling depth and sequencing technology error rates on rarefaction, I would try to find a more recent citation for this range or remove it.

Line 67: correct to “parts”

Line 68: correct to “genomes”

Line 68: the use of a comma-delineated list confuses meaning here, as it looks like “microbiome” is another thing the human gut hosts and not just a clarification for “microorganisms and their genome”.  I suggest using “i.e.” or parenthesis to distinguish “microbiome”

Line 70: the authors previously referred to the microbiome within the gut, so the reader is going to assume all mentions of “the microbiome” refer to the one in the gut, unless the authors specify “the human microbiome”.  Thus, the phrase “most of the microbiome is located in the gastrointestinal tract” is confusing and redundant

Line 71: are “xeno-metabolome” and “micrometabolome” not the same thing?

Line 74: “large world…” could be corrected to “global-collaborative genetic projects”

Line 77: correct to “ a large amount of data that has allowed us to gain”

Line 79: “Archaea”

Line 80: provide mention of typical intestinal fungi and archaea?  Could probably say that intestinal fungal populations are often highly diet-specific.  Most studies that report on archaea in the human GI tract don’t use primers that can detect sufficient 16S resolution for archaea, and will report only that “Methanobrevibacter” were found, but within this one genus, the species’ composition will be dependent on diet and health status: Ishaq et al. 2016: https://cdn.intechopen.com/pdfs/51799.pdf . Even if fungal and archaeal populations in disease status are only incidentally changed, and not involved in disease pathology, they still participate in an altered community function in the gut

 Line 95: while all of these factors influence the gut microbiome, they do so in different magnitudes and durations, and diet has a stronger influence than birth mode. I would recommend changing the order of this list (even though these concepts are later explored in that order), because the proximity of “mainly” and “birth mode” gives the impression that this is the most important factor. Further, mode of delivery does affect the gut microbiome, but this is shown to be overridden by gestational age (Korpela et al. 2018 https://www.nature.com/articles/s41598-018-20827-x) and re-colonization using body-site (Dominguez-Bello et al. 2016 https://www.ncbi.nlm.nih.gov/pmc/articles/PMC5062956/ ).  Further, babies delivered by C-section are exposed to antibiotics (ex. Stokholm et al. 2018 https://www.sciencedirect.com/science/article/pii/S0091674916002967 ) or are delivered because of medical complications and are in physiological distress, so many of these studies don’t really present two comparable cohorts.

Line 111-112: there have also been several studies identifying bacteria naturally found in the breastmilk of humans and other mammals that have a high-similarity to those later found in the infant gut (discussed in Yeoman et al. 2018 https://www.nature.com/articles/s41598-018-21440-8 )

Line 122: “most influences” is vague enough to be misleading, “most dramatically remodels in the short term” might be better, especially as the following sentence mentions recovery

Line 124 – 128: these sentences conceptually seem out of order, I would recommend reordering them, unless the authors are going to make a statement about how perinatal antibiotic use leads to different community reassembly dynamics because a stable microbiota have not yet been established.

Line 135: yeasts are also commonly used in probiotics

Line 169: “, is the dominant one in breast-fed newborns” is vague, do the authors mean “, is the dominant species of Bifidobacterium in the gut of breast-fed newborns”?

Line 170” correct to “firstly”

Line 176: create a new paragraph starting here

Line 178: correct to “to adaptation and competition”

Line 234: correct to: “it was suitable”

Line 236: “demonstrated”

Line 239: correct to “included in a commercial high”

Line 241: “components”

Line 250: “observations”

Line 293: “partial”

Line 296: should “determined” be “demonstrated”?

Line 307: correct to “oral”

Line 312 – 354: While these paragraphs provide interesting information for B. breve, it is not relevant to the review in that it is not related to pediatric diseases.  I suggest moving it to section 7 on adults.

Line 357: “Being that bifidobacteria is the most”

Line 359: should “subjected to” be “susceptible to”?

Line 407: “week”

Line 419: “consisted of”

Line 421: correct “resulted” to “was”

Line 472: correct to “in children becoming overweight”?

Line 476-477: “related to the part previous the cross-over” not sure what this is trying to say

Line 481: correct “less” to “fewer”

Line 487: “significant”

Line 509: “has been demonstrated”

Line 513: correct “which an infectious” to “which an infection”

Line 542: no years are given (although reference numbers are listed) in table 1 so the point about chronological order is somewhat irrelevant, especially since the citations only date back to 2012.

Line 555: “began”

Line 557: “mixed with water”

Line 575: “mixed with human milk”

Author Response

Response to Reviewer 2 Comments

Reviewer 2) Overall, the review is very thorough and it is clear that the authors put an immense amount of work into compiling this.  However, this also makes the review very dense.  A few paragraphs which do not touch on pediatric issues could be removed or reduced. 

Response:  In agreement also with the suggestions of the other reviewer, we reduced the paragraphs 1, 2, 3 and 4 in order to mainly emphasize the use of B. breve in paediatrics (overall, about 1 page reduction).

It is also very difficult to glean a thorough understanding of the potential for Bifidobacterium pertaining to different body systems or disease etiologies, as the information on any one area (ex. GI diseases) treated by probiotics is scattered across several sections.  I would suggest reorganizing sections (starting from line 162) into disease type and applications of Bifidobacterium, ex. “gastrointestinaldisorders”, “nutritional status and development”, “respiratory infections”, “allergies and immune system development”.  This would aid the reader in appreciating the amount of work that has been done on Bifidobacterium as a probiotic, and dramatically improve the readability of the manuscript which is currently disjointed.  While this will require a major revision of the text layout, it will not require a major revision of the scope or content of the manuscript. 

Response: As suggested, section 5 and 6 have re-organized into disease type-section.

 Line 12: correct to “for human health”

Response: We provided for the correction

Line 13: “gut microbiota”?

Response: As the word “dysbiosis” is referred to a general microorganism imbalance we omitted the term “microbiota” leaving only the term “gut” but if the reviewer believes that the general meaning results not clear we will provide for the correction. However for better clarity, we added the word.

Line 14: “not apparently liked to the gut environment” is a bit vague, as I assume the authors mean “not linked directly to host biology”?

Response : Actually we mean “not apparently linked to the gastrointestinal tract”, because we are focusing on gut microbiota  and its involvement in specific diseases; a dysbiosis is mostly associated to gastrointestinal disorders, but several studies, also showed in this review, have demonstrated that an imbalanced assessment of gut microbiota could contribute to the pathogenesis of “non-enteric” diseases, as allergies, autoimmune diseases and neurological diseases. Therefore with that sentence we wanted to underline the important role of gut microbiota in maintaining the overall normal human physiology, since alteration of gut microbiota composition can contribute to the occurrence of not only enteric diseases but also disorders that affect other districts. However, for better clarity, we changed the sentence in “not apparently linked to the gastrointestinal tract”.

Line 37: remove comma after species 

 Response: We provided for the correction

Line 55: I don’t think the 15,000 – 36,000 species estimate is accurate, and at best the original citation is probably pointing to sequence variants or strains.  That calculation is based on two clone-library studies and one early sequencing study, the highest of which estimated about 4,000 bacterial phylotypes, and all three studies use a relatively small library from which to estimate rarefaction curves.  Given the effect of sampling depth and sequencing technology error rates on rarefaction, I would try to find a more recent citation for this range or remove it.

Response: Thank you for the observation. In order to avoid ambiguities we have reported the explanation for the range values reported in the text

Line 67: correct to “parts”

Response: We provided for the correction

Line 68: correct to “genomes”

 Response: We provided for the correction

Line 68: the use of a comma-delineated list confuses meaning here, as it looks like “microbiome” is another thing the human gut hosts and not just a clarification for “microorganisms and their genome”.  I suggest using “i.e.” or parenthesis to distinguish “microbiome”

 Response: Thank you for the suggestion, we provided for the clarification

Line 70: the authors previously referred to the microbiome within the gut, so the reader is going to assume all mentions of “the microbiome” refer to the one in the gut, unless the authors specify “the human microbiome”.  Thus, the phrase “most of the microbiome is located in the gastrointestinal tract” is confusing and redundant

  Response: Thank you for the suggestion, the sentence has been reorganized in order to better clarify the concept (lines 66-70)

Line 71: are “xeno-metabolome” and “micrometabolome” not the same thing?

Response: To the best of our knowledge,  both terms are referred to a set of metabolites but different for origins: “micrometabolome” is referred to the set of metabolites produced by the microbiota (e.g. vitamins and SCFAs constitute some micrometabolites of gut microbiota); “xeno-metabolome” refers to the set of metabolites derived from xeno-biotics that are chemical external substances (e.g. drugs, food additive, hydrocarbons, pesicides) found within a organism but not naturally produced or expected to be present in biological system. The human organism is able to remove xenobiotics performing a conversion to less toxic forms through the production of xeno-metabolites then excreted; in some cases the conversion process results in more toxic forms. This process, corresponding to a  bioactivation of these compounds,  can result in structural and functional changes to the gut microbiota (Park et al. 2011). Therefore, xeno-metabolome  is considered an external factor that can influence the intestinal microbiota community.

Line 74: “large world…” could be corrected to “global-collaborative genetic projects”

Response: We provided for the correction 

Line 77: correct to “ a large amount of data that has allowed us to gain”

 Response: We provided for the correction 

Line 79: “Archaea”

 Response: We provided for the correction 

Line 80: provide mention of typical intestinal fungi and archaea?  Could probably say that intestinal fungal populations are often highly diet-specific.  Most studies that report on archaea in the human GI tract don’t use primers that can detect sufficient 16S resolution for archaea, and will report only that “Methanobrevibacter” were found, but within this one genus, the species’ composition will be dependent on diet and health status: Ishaq et al. 2016: https://cdn.intechopen.com/pdfs/51799.pdf . Even if fungal and archaeal populations in disease status are only incidentally changed, and not involved in disease pathology, they still participate in an altered community function in the gut

Response: Thank you for the suggestion, we provided for the insertion of this concept (please see lines 79-85)

Round  2

Reviewer 1 Report

The article now is cleraly presented. I have no more suggestions.

Author Response

Thank you for appreciating our revision work.

Reviewer 2 Report

The authors have done an excellent job addressing review comments, and the paper is much improved.  Only two spelling corrections are recommended.

line 108: table should be "stable"

line 770: correct to "use of B." (delete the t)

Author Response

Dear Reviewer,

thank you for appreciating our revision work. We have provided the spelling corrections you asked for.

Kind regards,

Diana Di Gioia